# Pharmacist-Led Deprescribing of Opioids and Benzodiazepines in Older Adults: Examining Implementation and Perceptions

**DOI:** 10.3390/pharmacy12040119

**Published:** 2024-07-30

**Authors:** Tamera D. Hughes, Elizabeth Sottung, Juliet Nowak, Kimberly A. Sanders

**Affiliations:** Eshelman School of Pharmacy, University of North Carolina at Chapel Hill, Chapel Hill, NC 27599, USAkim.sanders@unc.edu (K.A.S.)

**Keywords:** deprescribing, pharmacist-led intervention, older adults, opioids, benzodiazepines

## Abstract

**Background:** This study examines the implementation and perceptions of a pharmacist consultant deprescribing program aimed at reducing the risk of falls in older adults using opioids and benzodiazepines. **Methods:** This qualitative study conducted interviews with healthcare providers. The interviews were conducted from August to December 2021 and analyzed using inductive coding techniques. **Results:** Five participants, predominantly female MDs or PA-Cs from rural clinics, were interviewed. The participants adopted a pharmacist-led deprescribing program due to their heightened awareness of the opioid crisis, dedication to patient safety, and a desire for opioid deprescribing education. Initially, concerns included patient resistance and provider-driven barriers. However, over time, patient attitudes shifted toward greater openness to the program. The providers emphasized several critical needs for the success of the program: guaranteed access to pharmacists, tailored patient education, resources specific to providers, and financial support, including telehealth options. These factors were deemed essential to overcoming initial barriers and ensuring effective implementation. **Conclusion:** Integrating pharmacists into primary care settings shows promise for deprescribing opioids and benzodiazepines in older adults. Future research should explore telehealth options for patient–pharmacist consultations and expand the application of these findings to other healthcare settings. The study highlights the importance of awareness, patient education, access to resources (pharmacists), and provider support in addressing deprescribing among older adults.

## 1. Introduction

Opioids and benzodiazepines (BZDs) pose high risks of medication-related adverse side effects for older adults, including cognitive impairment, delirium, fractures, and falls [1,2]. Despite these risks, recent data indicate that benzodiazepine and opioid prescriptions for adults aged 65 and older remain high [2,3]. A 2019 study found a doubling of primary care office visits nationally between 2003 and 2015 in which a benzodiazepine was prescribed (3.1% to 6.4%) [4]. Between 2010 and 2015, opioid-related hospitalizations increased by 34% and emergency department visits increased by 74% for adults aged 65 and older [5]. Thus, the importance of deprescribing BZDs and opioids in older adults is becoming more crucial among providers in clinical practice.

Researchers at the University of North Carolina (UNC) introduced a pharmacist consultant deprescribing program. This program integrated targeted consultant pharmacists into a collaborative care model to reduce the fall risk in older adults overusing opioids and benzodiazepines [6,7]. With the targeted consultant pharmacist services, clinical pharmacists who were not embedded in the study’s outpatient clinics offered recommendations for deprescribing opioids and BZDs to clinic-based providers. As part of the study, pharmacists used electronic health records (EHRs) to inform providers at outpatient clinics (primary and specialty care) of older adults who were considered high risk for falls due to the chronic use of opioids and/or BZDs. Once alerted, the clinics used a team-based approach to screen and plan fall prevention interventions focused on deprescribing. Additional details on the consultant pharmacists intervention can be found in the literature [7,8].

This present study examines the implementation and perceptions of this pharmacist consultant deprescribing program. It identifies facilitators and challenges in integrating this program into primary care clinics and describes primary care clinicians’ views on the role of pharmacists in deprescribing opioids and benzodiazepines for older adults. The findings aim to provide insights into the implementation process and clinician attitudes, potentially informing the development of targeted interventions to support deprescribing efforts in this population.

## 2. Materials and Methods

This qualitative study was part of a larger pragmatic randomized trial (registered at clinicaltrials.gov: NCT04272671) investigating the effectiveness of a clinical pharmacist intervention for deprescribing opioids and BZDs in older adults at risk for falls. The study was conducted according to the Declaration of Helsinki guidelines and approved by the Institutional Review Board of The University of North Carolina at Chapel Hill. The larger trial’s goal was to investigate the effectiveness of a clinical pharmacist intervention to identify individuals aged ≥65 years at risk for falls based on chronic opioid and/or BZD usage. Subsequently, the intervention aimed to provide specific deprescribing recommendations to primary care providers for these individuals. The trial’s primary care practices were part of a physicians’ network academic healthcare system in North Carolina, which included over 90 outpatient primary care facilities spread across 14 counties. Ten clinics in rural regions and five in suburban or metropolitan counties consented to take part in the study. This present study begins at the conclusion of the overall trial’s intervention.

### 2.1. Study Design and Setting

This qualitative study used a semi-structured interview to capture participant experiences with implementing the pharmacist consultant model into their practice setting. Participants were identified through purposeful sampling and invited by email to take part in an interview. After two emails with no response, the researchers did not pursue the participants further. Interviews were conducted from August to December 2021 by one member of the research team, TH, via Zoom Client for Meetings [computer program] Version 4.6.7. (Zoom Video Communications Inc., San Jose, CA, USA). TH served as the implementation facilitation coach for the overall study and frequently engaged with the study participants throughout the intervention process. Detailed information about this relationship can be found in the study’s protocol [7]. TH conducted the interviews in her workplace. TH collected verbal informed consent before conducting the interviews. Each interview was approximately 60 min and was audio recorded and transcribed verbatim (ES and JN).

### 2.2. Data Collection

The semi-structured interview was guided by the three clusters of influence on the rate of diffusion of innovations within healthcare organizations. The three clusters include identifying the characteristics of the individuals who adopt the change, the perceptions of the innovation, and the contextual and managerial factors within each organization [9]. Additionally, questions focused on specific areas of the intervention, including program resources and experiences with patients, staff, pharmacists, and the research team (Appendix A). The interview questions were written by one researcher and then discussed with the research team from the larger study to ensure the clarity and relevance of the questions. Data collection concluded upon reaching saturation for the interviews [10]. Saturation, as defined within the study, denotes the stage at which no novel information emerged from the collected data. This criterion ensured that the research captured a comprehensive understanding of the subject matter, thereby enhancing the validity and reliability of the findings.

### 2.3. Data Analysis

To reflect the opinions of the participants, codes were developed from the data using an inductive or open-coding technique [11,12]. The entire study team (TH, ES, and JN) iterated through the first coding phase, reading and freely discussing the participant replies. From these discussions, codes and related definitions were developed. Memos were produced before, during, and following each coding session to document the analysis process and any developing themes or patterns in the data [11,12]. Following the initial data coding, participants’ main thoughts were identified by iteratively reviewing the data using individual summary matrices built on the interview questions. Cluster analysis was used to focus on the data that generated significant concepts that were presented by the participants [11,12]. From the cluster analysis, themes were identified. Microsoft Word and Excel were used to organize and analyze the data. All of the analysis was completed at the researchers’ workplace/university. The reporting of this study is in line with the consolidated criteria for reporting qualitative research (COREQ) [13].

## 3. Results

For each of the three clusters of influence, our analysis revealed the following themes:

### 3.1. Characteristics of the Individuals Who Adopt the Change

Five participants responded and agreed to be interviewed. Table 1 displays the characteristics of the individuals who agreed to participate in the interview, as well as the clinic demographics from which they participated. Most of the participants were female and held either an MD or PA-C. All the participants were from clinics in rural areas.

#### 3.1.1. Perceptions of the Innovation

Most of the participants’ decisions to engage in the study centered around three themes: awareness of the opioid crisis, interests in protecting their patient population, and educating themselves on opioid deprescribing.

Many participants perceived that the intervention would bring attention to the opioid crisis at their clinic. One participant stated “*Since I’ve been manager, this is something that hasn’t been addressed. It was a good opportunity to bring awareness*”.

Most of the participants saw an opportunity to help their general practice population by supplementing their access to resources and supporting the move towards collaborating care with pharmacists. One participant stated, “*This [intervention] seemed so high yield and when you have an [older adult] population, we can take better care of them*”.

Others perceived the intervention as educational, having admitted to not knowing how to go about deprescribing, stating, “*I think [participating in the intervention] was for educational purposes. Because of the opioid crisis, and the changes and practices… people are trying to get people off the medications if it’s no longer appropriate*”.

Respondents’ willingness to participate in the intervention was accompanied by some reservations about how it might affect their practice, including patient-driven concerns such as a lack of understanding and unwillingness to change. Participants felt that their patients, especially older adults, would have a hard time understanding why their medications were being deprescribed. When asked about their concerns prior to the intervention, one participant said, “*Backlash from patients and just the frustration. The ‘why now’ type question. It’s been working well for me. [Patients] just not understanding the long-term effects*”.

Several participants mentioned how their patients were first unwilling to deprescribe but came to rethink this as the study continued. One participant said, “*To start off they were not accepting of any change. During the study, it became possible to consider change. It went from no to let me think about it*”.

The participants not only had concerns about the patient’s willingness to participate, but also about their abilities to properly deprescribe. Some provider-driven concerns that were identified were a lack of time and stress. Participants felt that their days were already packed with appointments, and adding another topic to discuss might not fit into patient conversations. Participants were aware that if this were to be done correctly, time was needed to fully explain the deprescribing process and why it is necessary. When asked about their concerns prior to the intervention, one participant stated, “*How much time is this going to take, and how many meetings do we have to go to*?”

Another said, “*Time. The demands. I feel that we spend a lot of time doing clerical things instead of medical things. I was worried about how much additional clerical time was going to be required*”.

Participants also considered the potential liability consequences of participating in the study. One participant revealed, “*Some [providers] worry that there’s some liability if you’re not doing what the pharmacist says is recommended*”.

Notwithstanding these apprehensions, respondents shared a collective sense that participating in the research study held the potential for considerable benefits to both their patients and their practice.

#### 3.1.2. Contextual and Managerial Factors within Each Organization

Prior to the intervention, many of the contextual and managerial characteristics inside the intervention clinics suggested no culture of deprescribing or determining fall risk. Participants discussed having an informal process to identify patients at risk for falls. “*There was no preventative culture. It was an informal process where the nurse manager would ask the STEADI questions regarding fall risk*,” stated one participant.

Participants also discussed patient defensiveness when elaborating on the patient’s response to presenting the pharmacist consultant model. “*The patients at first were not happy. They were defensive*”.

Furthermore, participants agreed that it was very difficult to start conversations regarding deprescribing. There was hesitancy, and some participants did not feel as comfortable when they did not see it was appropriate. “*[Prior to the intervention] Some providers were hesitant and would relieve themselves of this duty if they were not comfortable starting deprescribing conversations*?” stated one participant.

Another participant said, “*It was unusual to have a patient respond in a positive way when the discussion started. It was very difficult at first, but it got easier each time*”.

With the intervention complete, most of the participants discussed a willingness and an openness to sustain and/or create a culture around deprescribing. One participant responded, “*With the pharmacist involved, the deprescribing intervention was better. It was another set of eyes*”.

However, the participants also discussed that this culture would be an ongoing process that would need improvements throughout the progression of the intervention in other clinics. “*The pharmacist input was very helpful, however it’s important to encourage openness from providers to accept pharmacists into clinics. There needs to be a partnership between providers and pharmacists*?” stated one participant.

Nevertheless, all participants agreed that repetition led to more conversation and, in turn, enhanced the culture surrounding deprescribing in their practice. “*We used repetition, for patients who were open to the idea of deprescribing. It’s an evolving process where a strong patient/provider relationship is essential to build trust which opens communication and breaks barriers that were not established prior*”.

It did not appear, however, that having clinic champions as part of the implementation process affected how well the participants in the clinic took up the services. “*I brought the opportunity to us, as clinic champion, and the clinic agreed it would be a great opportunity for us. I talked with the providers beforehand*”*,* said a study participant.

Participants who did take on a champion role, however, appeared to be more engaged in the process. One participant stated, “*It was solely my own decision to participate in the study. As the clinic champion, it was important to keep the clinic staff engaged, by discussing the patients prior to seeing the patients that day, to prepare for what would be discussed later on*”.

#### 3.1.3. Needs to Implement Pharmacist Consultant Model in the Future

The interviewed participants/providers emphasized the importance of several key elements for future implementation: guaranteed access to pharmacists, patient-specific educational materials, provider-specific educational resources, and financial support. They also considered potential strategies, such as having pharmacists play a more extensive role in the deprescribing process, either in person, virtually, or through telehealth.

One participant stated needing “*A pharmacist, the most labor intensive would be someone who would screen our patients remotely for use of benzodiazepines or narcotics, the less labor intensive version would be for us to a have a list of all our patients on narcotics and benzodiazepines and have the pharmacist sort of do recommendations that would be sent to us*”.

The present study also found that participants felt that not enough information was given to the patients, and there was a need to include printed materials to further support and educate patients, including the conversation starter that was a part of intervention materials [7].

Concerns about sustaining the proposed intervention once extramural funding had ended also emerged. One participant explained: “*Financially, I don’t believe it’s costing our clinic a thing. It might eventually if [pharmacists] were officially on staff*”.

## 4. Discussion

This qualitative study serves to cover the perspectives of providers in UNC physician network clinics tasked with implementing a novel pharmacist-led deprescribing study. The themes identified in this study reveal insights into the characteristics of the individuals who are adopting the change*,* perceptions of the innovation, and contextual considerations to enhance pharmacist-led deprescribing implementation in primary care settings.


*Perceptions of the innovation*


According to the diffusion of innovation, innovation characteristics explain 49–87% of the rate of adoption of innovations [14]. Most of the participants in this study decided to adopt the intervention owing to greater awareness of the opioid crisis, interests in protecting their patient population, and educating themselves on opioid deprescribing.


*Awareness*


Awareness is key to a lot of decision making in healthcare. Some studies have shown that physician awareness leads to a better understanding of patient outcomes [15,16]. When it comes to preventing opioid misuse, the U.S. Department of Health and Human Services has expressed the need for healthcare providers to be fully aware of inappropriate opioid use [17]. In our study, we found that provider awareness of the opioid crisis led to a greater longing to learn about deprescribing opioids and benzodiazepines in older adults. This is not the first time that provider awareness of a potential harm to patients has been integral in the deprescribing process. Sheehan et al. [18] also discusses physicians wanting to increase their knowledge surrounding deprescribing in older adults.


*Patient population*


Providers also discussed wanting to participate in this study because they dealt with older adults. As this population continues to age, there is significant focus on the outcomes of older adults. Considering that opioids and BZDs lead to poorer outcomes in this population, it begs to reason that providers would be interested in learning how to mitigate these effects through deprescribing. Similarly, other studies have also explored interventions for primary care practices and deprescribing in older adults [18]. The TAILOR study aimed to enhance the comprehension of the most effective methods for facilitating deprescribing among older individuals dealing with multimorbidity and polypharmacy [19]. Another study investigated the experiences and viewpoints of patients and providers regarding a rare deprescribing intervention in hemodialysis [20]. We foresee more primary care clinics getting involved with new interventions aimed at deprescribing to improve older adults’ outcomes.


*Education*


The continual education of healthcare providers is critical in our constantly evolving healthcare industry. Furthermore, the need to quickly and accurately apply research results to practice is growing in importance, as clinical research advancements quickly transform how medical care is delivered. In fact, one of the reasons physicians choose to participate in clinical trials or research is for educational purposes [21]. A study from 2012 investigated that providers participate in clinical trials for the following motives: altruistic reasons, self-interest of one’s reputation, and quality-of-care benefits. Quality-of-care benefits were noted across all sites, which meant the physicians were able to stay current with new advances in therapy and provide opportunities of education. We noticed the same response from our participants in the pharmacist deprescribing intervention. Many of the physicians learned how to properly deprescribe patients off opioids and BZDs and felt more comfortable doing so after the intervention.


*Barriers to deprescribing*


The literature reports extensively on barriers to deprescribing [22,23,24,25,26,27]. These barriers have included both patient-driven and provider-driven concerns. Therefore, it was not surprising that our participants also discussed these same concerns regarding participating in the study. In the case of opioids, deprescribing can be particularly challenging due to a variety of barriers that healthcare providers and patients encounter. Healthcare providers often grapple with their own concerns when it comes to deprescribing opioids. These concerns may include worries about patient satisfaction, the potential for patient withdrawal symptoms, and the fear of exacerbating pain. Such concerns can result in providers prescribing opioids longer than medically necessary. Another significant barrier is the lack of clear deprescribing guidelines specific to opioids or BZDs in older adults. The absence of standardized guidelines can leave healthcare providers unsure about how to proceed, contributing to inertia in deprescribing efforts [28,29]. The participant responses in this study hint that pharmacist integration in deprescribing may be the key to overcome many of these barriers. Similarly, Jordan et al. found that implementing pharmacist-led coordinated stewardship facilitated improved opioid management [30].


*Provider-driven concerns*


Research is the core of evidence-based medical practice, and major clinical trials impact the ways certain providers run their practice [31]. While this is true, one of the main issues with running practice-based research trials is physician participation [32]. One study highlighted barriers as to why physicians choose to stay away from research, and those barriers happened to be present in our study. The number one reason was time involvement followed by resource issues and the lack of clinical and scientific rationale of research [33]. The examples listed under time involvement included extra research-related work and discussions with patients. While talking to our participants, we discovered that time and stress were two major concerns with participating in this intervention. Although these barriers were concerns at the beginning of the intervention, many of the participants did not find them to be issues afterwards.


*Patient-driven concerns*


Each practitioner has a unique set of difficulties when it comes to stopping the prescription of opioids and BZDs to older adults. Our earlier research identified issues for particular patient demographics, including those who obtained prescription medications from other doctors, patients who used larger doses of opioids and BZDs, and patients who lacked access to mental health facilities for dealing with alcohol/substance misuse [34]. Our current study demonstrates that healthcare professionals also notice deprescribing obstacles in individuals who are blatantly opposed to any prescription adjustments and lack awareness of the opioid issue.


*Contextual and managerial factors within each organization*


One important result that we found from our interviews was the ease of incorporating this new service into the providers’ routine. In one study trying to implement pharmacists in primary care to assist with diabetes support to enhance insulin prescribing, they found the intervention to not be successful due to the need for providers to change their routines to implement the services [35]. Our study, however, provides that providers reported minimal change in their routine. Being able to incorporate new services into routine activities and standardizing that approach is known to be a facilitating factor necessary for program implementation [36,37]. Not only were the providers able to incorporate the new intervention into their routine, but the study participants also discussed allowing individual providers to implement the new changes in the way they saw fit. This is important because studies have shown that providers struggle to balance the benefits of following strict protocols while also providing individualized care [38]. Therefore, we believe it is important to provide an intervention that easily adapts into a provider’s routine while also providing some degree of flexibility for healthcare decisions. This observation was also noted in the shifts in themes on culture around deprescribing prior to and after the intervention. Table 2 displays a summary of the differences noted between perspectives pre-intervention and perspectives post-intervention with the available supporting literature.


**Future needs for implementation**


The present study, like others, has found that primary care clinics enjoy pharmacists being a part of the team [42,43]. We found that the providers were very accepting of the pharmacists’ recommendations. So much so that they explored the possibility of having pharmacists participate in an actual part of the dialogue with patients. This further adds to the literature, as prior studies suggest a large range in the acceptance of pharmacist recommendations [44]. Concerning the mode of intervention deliver (including a pharmacist in the clinic), we feel that further attention is needed, since economic analyses are frequently required to evaluate whether this arrangement is cost effective for clinics. On the other hand, we present an alternative to in-clinic pharmacists and demonstrate that while primary care providers would prefer a pharmacist in the clinic, they are generally agreeable on receiving pharmacist recommendations through telehealth [45,46]. Future research, on the other hand, might include telehealth appointments with pharmacists and patients to conduct deprescribing discussions.

The adaptation of telepharmacy in healthcare has drastically changed the way patients are treated and managed. A study from 2018 researched the implementation of a physician–pharmacist collaborative practice for the management of hypertension and revealed great benefits of telehealth in terms of patient outcomes and medication optimization [47]. Similarly, this intervention was centered around a multidisciplinary team involving telehealth pharmacists to deliver patient-specific deprescribing protocols. The intervention was a success in terms of deprescribing opioids and BZDs as well as increasing education of the opioid crisis in clinics.


**Limitations**


This study has limitations in terms of generalizability due to the small sample size and regional location of the participants. These factors highlight the need for further research with a larger and more diverse sample. Of note, data saturation was achieved with the participation of five individuals who responded and agreed to be interviewed, suggesting that the information gathered can be applied to future initiatives where pharmacists are integrated into primary care practices under similar circumstances. In this case, the participants were from the same regional locations, and all had a vested interest in deprescribing opioids and BZDs in their patient population. Furthermore, this study was based on a theoretical framework, which has been extensively studied in the literature to characterize the perceptions of innovating new services [9].

## 5. Conclusions

The escalating crisis of opioid and benzodiazepine overuse among older adults demands immediate attention and innovative solutions within primary care settings. This study sheds light on the challenges and opportunities associated with participating in a pharmacist consultant deprescribing program. Our findings underscore the challenges and opportunities associated with pharmacist consultant deprescribing programs, shedding light on a healthcare landscape in flux, where awareness of the opioid crisis, dedication to patient well-being, and a hunger for education serve as catalysts for change. While our study focused on rural clinics, future research should explore these differences to develop targeted interventions suitable for diverse healthcare environments, including non-rural areas.

## Figures and Tables

**Table 1 pharmacy-12-00119-t001:** Participant and clinic demographics.

	Deprescribing Experiences in Years	Clinic Population 65+	Clinic Volume Opioids	Clinic Volume BZDs	Total Providers	% MD Providers
**Participant 1**	8	1583	75 (4.7%)	54 (3.4%)	8	62.5%
**Participant 2**	0	1321	26 (2.0%)	31 (2.3%)	3	33.3%
**Participant 3**	20+	827	34 (4.1%)	46 (5.5%)	2	50%
**Participant 4**	23	1583	75 (4.7%)	54 (3.4%)	8	62.5%
**Participant 5**	5	827	34 (4.1%)	46 (5.5%)	2	50%

**Table 2 pharmacy-12-00119-t002:** Summary of theme shifts on culture around deprescribing prior to and after the intervention.

Pre-Intervention Issues	Participant Concerns	Post-Intervention Response and Additional Supporting Literature
**Informal Process**	- Participants reported a lack of established practice processes - Described an informal approach - Indicated asking STEADI questions for screening but lacked comprehensive understanding of rationale	- Open to change to sustain and/or create a culture of deprescribing - Presence of pharmacy in clinical teams can result in more suitable medication usage and reduced adverse drug events, and a significant portion of patients expressed confidence in pharmacists overseeing their medications [39]
**Ongoing/Evolving Process**	- Emphasis was mainly on discontinuation of specific medications that are problematic in older adults	- Participants portrayed deprescribing as a continually evolving practice within their routines - Approach has transformed into a more comprehensive strategy known as “deprescribing” [39]
**Patient Defensiveness**	- Participants’ fear of inadequate pain management - Encountered challenges to engage in conversations about deprescribing from patients who are dependent on their opioid medications	- Positioning of pharmacy can support patients and providers throughout the process - Pharmacists have been taught to use motivational interviewing in their coursework, which makes them an ideal solution for overcoming this barrier [40]
**Difficult to initiate conversations**	- Challenges can include prescriber self-efficacy, challenges in effectively communicating the rationale behind deprescribing to patients, and patients’ awareness regarding the necessity for deprescribing [26,41] - Requires building a strong, trusting relationship with patients	- Repetition led to more conversation - Studies like the OPTIMIZE trial emphasized the importance of timing and repetition in delivering the intervention in assessing the impact of educating patients, their family members, and clinicians on deprescribing; the trial emphasized the importance of timing and repetition in delivering the intervention [40]

## Data Availability

The data presented in this study are available on request from the corresponding author due to privacy and ethical restrictions.

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
