# Peer review of "Pharmacist-Led Deprescribing of Opioids and Benzodiazepines in Older Adults: Examining Implementation and Perceptions"

_pharmacy, 2024, doi:10.3390/pharmacy12040119_

Round 1

Reviewer 1 Report

Comments and Suggestions for Authors

I read with interest the paper titled "Pharmacist-Led Deprescribing of Opioids and Benzodiazepines in Older Adults: Examining Implementation and Perceptions".

Abstract:

The authors based our paper in 5 interviews. This is a very small and poor sampling, which could conduct to biased conclusions. This is a limitation that should be adressed. 

Why rural clinics? Whats the expected difference for non-rural areas? Limitation in conclude, should be adressed.

The conclusions are too strong for a study that is based in 5 interviews. 

Introduction:

Data presented is based in 2015. Opioid crisis is well discussed during the last months, for sure new references can be found which represents better the reality of US. 

How much is the hospitalization rate in the last years?

References [6], [7] should appear as [6,7] and the point after the reference. Please check the instructions for authors. This should be extensively reviewed in the paper. 

Objective in the background differ from the objective in the abstract.

In abstract the objectives were "examine implementation and perception". In the background the objective is written as "identify challenges".

After the objective, authors are too subjective saying what the study, perphaps, will add. this is opinative and should be removed.

Material and Methods:

- No need to state the degrees of the research team. It is expected that the research team are the authors of the paper. 

- You stated your study as observational exploratory study conducted interviews - this should be stated as "qualitative study" or research.

- How was data saturation reached?

- How was sampling made?

- Which was the method of interview? Online or face-to-face

- Please add a table with code, cluster, subthemes and themes identified. 

- How were the codes defined? How do you generate initial codes and themes?

- Who reviewed the codes and themes? This should be extensively described in the end of the paper "Contribution of the authors"

- How were the codes clustered?

- Did you preform descriptive coding, interpretative coding and overarching themes analysis? Please describe. Or do you made iterative comparison? I would like to know more about the methods of coding. 

Results. 

Results mainly focus on "perceptions of the innovation", "contextual and managerial factors" and "needs for a consultant model in the future". From those results to the strong conclusion you expect to provide, its a long distance. From 5 interviews is hard to conclude anything with evidence. This is a major drawback of your paper. 

Limitations of the study shoudl be described further. 

Author Response

Reviewer comment: The authors based our paper in 5 interviews. This is a very small and poor sampling, which could conduct to biased conclusions. This is a limitation that should be addressed.

Response: Thank you for your valuable feedback. We have addressed the limitations you pointed out in the manuscript. Specifically, we have acknowledged the small sample size of five interviews and the potential for biased conclusions due to the limited and regional participant pool.

Reviewer comment: Why rural clinics? What's the expected difference for non-rural areas? Limitation in conclusion should be addressed.

Response: We chose to focus on rural clinics due to their distinctive healthcare dynamics and often limited access to specialized services. However, the expected differences for non-rural areas warrant exploration to ensure the generalizability of our findings. Additionally, while our conclusions may appear strong, they should be tempered by the recognition of our study's limitations, particularly its small sample size.

Reviewer comment: The conclusions are too strong for a study that is based on 5 interviews.

Response: We have edited the conclusions.

Introduction:

Reviewer comment: Data presented is based in 2015. Opioid crisis is well discussed during the last months, for sure new references can be found which represents better the reality of US.

Response: Though the data is presented from 2015, notice that the studies were not published in 2020. We have slightly updated the introduction to reflect later studies; however, we decided to keep the current statistics as they highlight the literature that spurred the overall study’s objectives.

Reviewer comment: How much is the hospitalization rate in the last years?

Response: We currently do not have access to this information. Thank you for the suggestions.

Reviewer comment: References [6], [7] should appear as [6,7] and the point after the reference. Please check the instructions for authors. This should be extensively reviewed in the paper.

Response: References have been updated.

Reviewer comment: Objective in the background differ from the objective in the abstract. In abstract, the objectives were "examine implementation and perception". In the background, the objective is written as "identify challenges".

Response: This has been updated.

Reviewer comment: After the objective, authors are too subjective saying what the study, perhaps, will add. This is opinionative and should be removed.

Response: We decided to keep the aim but have added the word “potentially.”

Material and Methods:

Reviewer comment: No need to state the degrees of the research team. It is expected that the research team are the authors of the paper.

Response: We have removed the degrees. However, it is very common in qualitative writing to follow COREQ guidelines which list that authors do such in the methods.

Reviewer comment: You stated your study as observational exploratory study conducted interviews - this should be stated as "qualitative study" or research.

Response: We have updated this.

Reviewer comment: How was data saturation reached?

Response: We have added a section to discuss how data saturation was reached.

Reviewer comment: How was sampling made?

Response: Sampling is detailed in Study Design & Setting.

Reviewer comment: Which was the method of interview? Online or face-to-face

Response: It is mentioned in Study Design & Setting that interviews were conducted online over Zoom.

Reviewer comment: Please add a table with code, cluster, subthemes and themes identified.

Response: We do not believe this is necessary. However, if someone is interested, they may contact us via email to receive the files. We have made that invitation available in the Data Availability statement.

Reviewer comment: How were the codes defined? How do you generate initial codes and themes?

Response: This is defined in data analysis.

Reviewer comment: Who reviewed the codes and themes? This should be extensively described in the end of the paper "Contribution of the authors"

Response: This is defined in data analysis as well as at the end of the paper in the contribution of the authors. The initials also denote which team members did what.

Reviewer comment: How were the codes clustered?

Response: This is defined in data analysis.

Reviewer comment: Did you perform descriptive coding, interpretative coding and overarching themes analysis? Please describe. Or do you made iterative comparison? I would like to know more about the methods of coding.

Response: This is defined in data analysis.

Results:

Reviewer comment: Results mainly focus on "perceptions of the innovation", "contextual and managerial factors" and "needs for a consultant model in the future". From those results to the strong conclusion you expect to provide, it's a long distance. From 5 interviews is hard to conclude anything with evidence. This is a major drawback of your paper.

Response: We have updated this conclusion.

Reviewer comment: Limitations of the study should be described further.

Response: We have expanded the limitations.

Reviewer 2 Report

Comments and Suggestions for Authors

• When stating: "This observational, exploratory study was conducted by a research team consisting of TH, PharmD, PhD, RPh, and two pharmacy students, ES, PharmDc, and JN, PharmDc," what is the relevance of mentioning the gender of the researchers? I suggest omitting this information at this juncture, as it can be readily inferred from the authors' names on the manuscript.

• In line with the open science policies advocated by the journal and the publisher, I suggest making available to readers, either as an appendix or in a repository, the materials used for data collection and extraction, such as survey formats, databases, and any other materials employed, while ensuring the confidentiality of participant identities.

• I inquire whether 5 subjects might be too few to fulfill the study objectives. Kindly provide an explanation and justification.

• Regarding Table 1, I am unclear about the variable "clinical volume." I believe it is important to include some sociodemographic data of the interviewed subjects in this table, such as age, gender, specific postgraduate training, and professional experience, while safeguarding participant anonymity.

Author Response

  • When stating: "This observational, exploratory study was conducted by a research team consisting of TH, PharmD, PhD, RPh, and two pharmacy students, ES, PharmDc, and JN, PharmDc," what is the relevance of mentioning the gender of the researchers? I suggest omitting this information at this juncture, as it can be readily inferred from the authors' names on the manuscript.
  • We have removed this information. We were reporting results according to COREQ, but have not removed. Thank you.

In line with the open science policies advocated by the journal and the publisher, I suggest making available to readers, either as an appendix or in a repository, the materials used for data collection and extraction, such as survey formats, databases, and any other materials employed, while ensuring the confidentiality of participant identities.

  •  
  • We are open to allow for the interview questions to be provided supplementary and will add that information. However, as this study was in partnership with a government agency, we would have to receive their approval before sending over this information. However we have included information in the Data Availability Statement to inform readers that they may receive data upon request to the corresponding authors.

  • I inquire whether 5 subjects might be too few to fulfill the study objectives. Kindly provide an explanation and justification.
  • We have added additional information to show how data saturation was achieved. Data saturation was achieved at 5 participants.

    Regarding Table 1, I am unclear about the variable "clinical volume." I believe it is important to include some sociodemographic data of the interviewed subjects in this table, such as age, gender, specific postgraduate training, and professional experience, while safeguarding participant anonymity.
  • Clinic volume is necessary because it lets us know how “busy” these clinics are. Volume is highly linked to the capacity in which clinics are able to take on new services. Also, given the rural nature of these environments, we do not believe it necessary to view other demographic natures of the participants.

Reviewer 3 Report

Comments and Suggestions for Authors

The article "Pharmacist-Led Deprescribing of Opioids and Benzodiazepines 2 in Older Adults: Examining Implementation and Perceptions" by Hughes et al, is an interesting and well written manuscript.

However, there are some points I would like to address:

1. The results part of the abstract is difficult to follow and should be explained in some more detail.

2. There were five participants in the study, but how many participants did you want to include? Is there any calculation of the number of participants needed? 

3. In the results part you use the wording many participants, most of the participants, other.... Since there are only five participants I think you should say precisely how many participants you refer to. Moreover, what is the difference between participant and provider? Were there more providers than participants, it should be explained in more detail.

4. In the discussion, line 298  a reference is needed at the end of the sentence 

Author Response

  1. The results part of the abstract is difficult to follow and should be explained in some more detail.
    1. Thank you for this comment. We have updated the abstract.
  2. There were five participants in the study, but how many participants did you want to include? Is there any calculation of the number of participants needed? 
    1. We did not have a specified amount of participants we wanted to include. Also, in qualitative methods, there is no calculation to determine the number of participants needed. This is identified through data saturation, which was achieved in this study. It just so happened that we were able to achieve data saturation, meaning no new ideas were introduced, at 5.
  3. In the results part you use the wording many participants, most of the participants, other.... Since there are only five participants I think you should say precisely how many participants you refer to. Moreover, what is the difference between participant and provider? Were there more providers than participants, it should be explained in more detail.
    1. All the participants were providers. We have updated this in the manuscript to reduce confusion. Regarding your suggestion to specify the number of participants referred to with terms like "many" or "most," I would like to clarify that in qualitative research, the emphasis is typically on thematic saturation and depth of insights rather than on numerical representation, which is more characteristic of quantitative analysis. The use of terms such as "many" or "most" in qualitative studies aims to convey the prevalence of themes or perceptions among the participants without implying statistical significance.
  4. In the discussion, line 298  a reference is needed at the end of the sentence.
    1. Can you clarify this line. Line 298 does not present in the discussion.

Reviewer 4 Report

Comments and Suggestions for Authors

Pharmacist-Led Deprescribing of Opioids and Benzodiazepines in Older Adults: Examining Implementation and Perceptions

In this manuscript, the authors explore the implementation and perceptions of a pharmacist consultant deprescribing program aimed at reducing falls in older adults using opioids and benzodiazepines. Conducted from August to December 2021, the qualitative study involved interviews with five healthcare providers, mostly female MDs or PA-Cs from rural clinics. Participants adopted the pharmacist-led program due to their awareness of the opioid crisis, dedication to patient safety, and desire for deprescribing education. Initial concerns about patient resistance and provider barriers shifted as patient attitudes became more open. Providers highlighted the need for guaranteed pharmacist access, patient-specific education, provider resources, and financial support, including telehealth options. The study shows promise in integrating pharmacists into primary care to deprescribe opioids and benzodiazepines in older adults. Future research should explore telehealth consultations and extend these findings to other healthcare settings, emphasizing awareness, patient education, resource access, and provider support.

 Comments:

The manuscript shows interesting insights although it describes a situation primarily American, especially due to the easy consumption of opioids. There is only one publication in the literature on this topic where some of the authors are the same as those of this manuscript. Reading the manuscript was difficult due to the numerous revisions and deleted sections within it. The conclusions are consistent with the evidence and the references are appropriate and exhaustive. The Tables provide detailed information and the limitations of this study are already considered by the authors. English language is fine.

The manuscript can be accepted in present form.

Author Response

no suggestions were made by the review to improve this manuscript

Round 2

Reviewer 1 Report

Comments and Suggestions for Authors

The improvements made in the paper does not provide enough scientific soundness for the paper to be accepted. My decision is to reject. Major limitations were already shared in the previous revision. Sample provided is very heterogeneous and the conclusions are tuned to be strong, based on a weak methodology. 

Author Response

No suggestions to change were made by the reviewer as they resolve to reject the manuscript.